



**Management-intensive Grazing Affects Soil Health**
Casey Shawver[1], James A. Ippolito[1*], Joe Brummer[1], Jason Ahola[2], and Ryan Rhoades[2]
[1]    Department of Soil and Crop Sciences, Colorado State University, Fort Collins 80523, USA
[2]    Department of Animal Sciences, Colorado State University, Fort Collins 80523, USA
*Corresponding author: Jim.Ippolito@colostate.edu
Keywords:  Management-intensive grazing; soil health; soil management assessment framework.
Highlights
1.    Soil health is affected by management-intensive grazing (MiG)
2.    Biological soil health index increased under MiG
3.    Nutrient and physical soil health indices decreased under MiG





**ABSTRACT**
Management-intensive Grazing (MiG) on irrigated, perennial pastures has steadily
increased in the western US due to pressure for reducing public lands grazing, overall declining
land available for pasture, and decreasing commodity prices. However, there are still many
unknowns regarding MiG and its environmental impact, especially with regards to soil health.
Over a two-year period, we studied changes in soil health under a full-scale, 82 ha pivot-irrigated
perennial pasture system grazed with ~ 230 animal units (AUs) using MiG. Soil analysis
included 11 soil characteristics aggregated into the Soil Management Assessment Framework
(SMAF), which outputs results for soil biological, physical, nutrient, chemical, and overall soil
health indices (SHI). Positive impacts were observed in the biological SHI due to increases in
microbial and enzymatic activities, even though soil organic C (SOC) remained relatively
unchanged; however, positive biological SHI changes are likely precursors to future SOC
increases. The nutrient SHI declined due to a reduction in plant-available soil P over time,
potentially due to greater plant uptake. A negative impact was also observed in the physical SHI,
driven primarily by increasing bulk density due to hoof pressure from cattle grazing. If managed
correctly, results suggest that irrigated, MiG systems have the potential for success with regards
to supporting grazing while promoting soil health for environmental and economic sustainability.



## 1. Introduction

Management-intensive Grazing (MiG) is often defined as "a flexible version of rotational grazing that balances forage supply with animal demand" (Stout et al., 2000). Over the past decade, interest in MiG on irrigated pastures has increased steadily due to the prospects of reduced production costs, increased animal output, land use efficiency, and environmental benefits. This system is being considered as an option by many farmers and ranchers in the western US due to pressure to reduce public land grazing and the declining land available for pasture (Cox et al., 2017). Adoption of MiG has the potential to bring the benefits of intensively managed, improved pastures into the already established irrigated cropping infrastructure that exists on many ranches. However, there are still many unknowns about the implications of an intensive cattle grazing system on a limited parcel of irrigated land, particularly in terms of soil health. One could imagine that studying changes in soil health (or more specifically the combination of soil biological, physical, and chemical attributes) would integrally improve our understanding of how MiG affects the overall function and viability of cropland converted to pasture systems.

Converting cropland to perennial pasture can enhance soil health by increasing microbial and enzymatic activity, building soil organic matter, and sequestering C (Acosta-Martínez et al., 2008; Carter et al., 1994; McCallum et al., 2004; Paudel et al., 2011; Veum et al., 2015). Acosta-Martinez et al. (2008) showed that microbial biomass C (MBC) was up to 6.6 times greater under pasture compared to corresponding vegetable production sites. Intimately correlated to microbial biomass C is the ability of microorganisms to degrade soil organic matter via enzymatic activity (Turner et al., 2002). Enzymes are integral to all soil biological activity, including organic matter decomposition. Specifically, the β-glucosidase (BG) enzyme plays an important role in the



process of cellulose degradation. As a group, the enzymes referred to as "glucosidases" release
significant energy sources in the form of sugars that are important for sustaining soil microbial
populations (Bandick and Dick, 1999). Bandick and Dick (1999) found that a permanent pasture
site had greater β-glucosidase activity and overall enzymatic activity as compared to several
agroecosystem production sites. Turner et al. (2002) found that β-glucosidase activity was
positively correlated to soil and microbial C concentrations under pasture soils. Soil microbial
and enzymatic activity are also likely linked to soil physical properties that affect soil water and
air relations in perennial systems.

Perennial pasture plants establish root systems that alter overall soil health by improving

physical properties such as aggregate stability, water infiltration, and sub-soil macroporosity
(Carter et al., 1994; McCallum et al., 2004; Milne and Haynes, 2004), all of which are likely
affected by improvements in both soil organic matter and microbiological activity. McCallum et
al. (2004) found that perennial pasture improved soil macroporosity and infiltration in the dense
B horizon of the soil profile, as well as increased the number of 0.3-0.03 mm pores. Adding
herbivory in perennial plant systems can further enhance these positive soil physical
improvements. Herbivory increases plant root exudation (Bardgett et al., 1998; Macduff and
Jackson, 1992; Shand et al., 2006), with root exudates playing a major role in binding soil
particles together to create greater aggregate stability. Aggregate stability can be further
enhanced via greater organic matter accumulation simply by eliminating tillage in perennial
systems (Tisdall and Oades, 1982). A study by Milne and Hayes (2004) supported this
contention, finding greater aggregate stability in perennial grazing systems compared to annual
ryegrass grazing systems.

Grazed pastures can also improve soil health by improving soil chemical and nutrient





properties. These systems tend to accumulate more potassium (K) and phosphorus (P) in the top
several cm of soil than systems where forage is hayed, simply due to manure inputs (Mathews et
al., 1994). When passing through the animal, approximately 96% of P is excreted in manure
(Eghball et al., 2002). Cattle manure P availability reaches and often exceeds 70%, primarily in
inorganic forms (Eghball et al., 2002). Availability of the remaining manure organic P is largely
controlled by microbial mineralization, which is influenced by soil temperature, moisture, and
manure characteristics such as animal species and diet (Eghball et al., 2002; Slavich and
Petterson, 1993). Approximately 73% of K is excreted by cattle in urine and is often 100%
bioavailable; this is not a large environmental concern in pasture systems (Eghball et al., 2002)
because K is generally considered immobile and only leaches in extreme cases of low soil pH
and cation exchange capacity. In a study determining the fate of potassium under irrigated
pasture, K losses through leaching were negligible (0.99 g m$^{-2}$ yr$^{-1}$; Early et al., 1998).

Effects from managed grazing are mostly positive. However, the addition of grazing,

particularly in irrigated systems, raises concerns about adverse effects on soil properties and thus
overall soil health. Specifically, increases in bulk density are of concern in managed perennial
pasture systems because of potential compaction caused by hoof-to-soil contact. Bulk density
increases can be exacerbated with increased soil moisture, higher stocking densities, and lower
amounts of surface litter within irrigated perennial pasture systems (Da Silva et al., 2003;
Drewry et al., 2008; Greenwood and McKenzie, 2001). Although certain methods have been
shown to decrease bulk density (e.g., aeration and deep-ripping; Greenwood et al., 1998;
Greenwood and Mckenzie, 2001; Malhi et al., 2011), natural recovery by removing grazing from
a system has been shown to return bulk density to levels comparable to ungrazed soils
(Greenwood et al., 1998). Da Silva et al. (2003) showed that decreasing amounts of post-graze

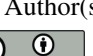



residue correlated to higher penetrometer resistance, an indirect measure of soil bulk density.
Positive and negative grazing impacts in perennial pasture systems can make or break the
viability of a livestock enterprise, with the primary driver behind yields, forage health, and
profitability being the ability of soils to function properly, the basic premise behind soil health.

To date, we are unaware of any studies focused on quantifying soil health in irrigated,

MiG systems. The overall study objective was to quantify soil health changes caused by a land-
use change from cropland to an irrigated, perennial pasture grazed by cattle using the Soil
Management Assessment Framework (SMAF), a soil health tool developed by Andrews et al.
(2004). Based on current literature, we hypothesized that converting irrigated cropland to
perennial, MiG pasture would cause 1) negative changes in the physical soil health index (SHI)
due to increases in bulk density exerted from hoof pressure; 2) the biological SHI to increase due
to microbial biomass C and enzymatic activity being stimulated from perennial grass roots and
lack of tillage; and 3) the nutrient SHI to increase due to greater P and K levels from manure and
urine deposition during the grazing season.



## 2. Materials and Methods

*2.1 Site Description*

This study was conducted at the Colorado State University Agricultural Research, Development and Education Center located 13 km northeast of Fort Collins, Colorado, USA (40°39'30.40" N, 104°59'11.24" W). The 82-ha research area was under center pivot irrigation. Climatic conditions were mid-latitude dry, cold, semi-arid steppe (Kottek et al., 2006), with average low and high temperatures of 1.0 to 16.8 °C, average annual rainfall of 380 mm (WRCC, 2018), and an elevation of 1,554 m. For the two-year study period (2017 and 2018), mean temperatures ranged from 6.9 to 22.7 °C and 10.7 to 27.5 °C, respectively (Colorado Climate Center, 2018). Irrigation totals for 2017 and 2018 were 45.6 and 58.9 cm, respectively. The following soils were within the study area: Connerton-Barnum complex (fine-loamy, mixed, mesic Torriorthentic Haplustoll and fine-loamy, mixed (calcareous), mesic Ustic Torrifluvent; 2 ha), wet Aquepts (5 ha), Garrett loam (fine-loamy, mixed, mesic Pachic Argiustoll; 7 ha), Kim loam (fine-loamy, mixed (calcareous), mesic Ustic Torriorthent; 10 ha), Otero sandy loam, (coarse-loamy, mixed (calcareous), mesic Aridic Ustorthent; 11 ha), and Nunn clay loam (fine, montmorillonitic, mesic Aridic Argiustoll; 48 ha).

Prior to project establishment, the study area was managed for about a decade as a tilled cropping system with crops including corn grain and silage (*Zea mays* L.), dry beans (*Phaseolus vulgaris* L.), and alfalfa (*Medicago sativa* L.). In 2016, the area was converted to four forage mixtures, one planted on each quarter (~ 20 ha) of the center pivot, including a simple grass-legume mix, complex grass-legume mix, simple grass mix, and complex grass mix (Table 1), with all grasses being cool-season. Prior to planting, a deep ripper was used to alleviate a plow pan that had formed due to the clayey soil texture that dominates the site, as well as previous



management. Following deep ripping, the field was moldboard plowed, disked twice,
cultipacked twice, and then rolled with a heavy steel roller to break up large soil aggregates and
firm the soil surface prior to seeding. Prior to the second cultipacking operation, 40 Rock
fertilizer (12-40-0-6.5 S-1 Zn, J.R. Simplot Company, Boise, ID) was broadcast at a bulk rate of
135 kg ha$^{-1}$ and immediately incorporated. Cool-season grasses were planted in late August and
early September 2016 and legumes were cross-drilled on March 15-16, 2017.
Unfortunate climatic circumstances in the spring of 2017 led to multiple issues in plant
establishment. Within the northeast quarter of the pivot, species mixture A (noted as A in Table
1) failed due to winds that moved loose soil and damaged small seedlings. Oats were planted in
this quarter on March 23, 2017 to avoid further wind erosion. The planned mixture was re-
planted in August 2017. For the mixtures containing legumes, most of the legumes that were
cross drilled in mid-March were killed due to a hard frost that occurred shortly after germination.
Establishment success of the legumes was approximated at less than 5%. Legumes were again
interseeded in early August of 2018. Thus, the forage mixtures with legumes contained few if
any legumes for the study duration.  Although plant species mixes were an important facet of the
overall project, for this manuscript we focus on the use of plant material removal percentage
within the context of MiG (see section 2.3, Grazing Management below).
*2.2  Project Design*
Permanent infrastructure included an electrified perimeter fence with two concentric
inner fences constructed using high-tensile wire (Fig. 1). Ten permanent water blocks were
located around the pivot, eight within the outer concentric high-tensile fence line and two in the
center. The eight water blocks within the outer concentric fence had four sides with electrified
rope gates for controlling access to paddocks. Paddocks were generally created every 1-3 days



using polywire and step-in posts. Fences that delineated paddocks were re-constructed in the
same locations throughout each grazing rotation using the GPS and paddock drawing tools on the
mobile application PastureMap™ ("PastureMap Grazing Management and Livestock Software,"
2019). Further subdivisions to these paddocks were also made on an as-needed basis using
additional polywire and step-in posts to adjust for forage availability and animal numbers. These
fences were not placed in the same location each grazing rotation.

In 2017, approximately 171 cow-calf pairs were grazed from August 18 until October 24.

The grazing season was delayed due to fencing and water infrastructure being constructed.
Because of the delay, initial growth of forage in mixtures B, C, and D was harvested for hay. A
total of 309 Mg of above-ground biomass was removed as hay, which equated to 5 Mg ha$^{-1}$.
When grazing commenced on August 18, the cows were initially separated into two herds by
breed (Angus and Hereford) for breeding purposes and then combined on September 21 and
grazed as one herd until October 24. In 2018, approximately 136 cow-calf pairs, 49 replacement
heifers, and 5 steers were grazed from May 4 to October 7, with similar animal separation for
breeding purposes as in 2017.
*2.3 Grazing Management*

MiG systems require manipulating the length of time cattle graze and space allotted

based on available forage resources to achieve management goals and objectives (Shewmaker
and Bohle, 2010). For this project, cows were generally moved daily. In certain situations,
depending on forage availability and herd size, cows were moved every 2-4 days. This
management method allowed for making daily adjustments in order to maintain electric fencing,
monitor cattle health and soil conditions, and preserve plant health.



In terms of plant health and performance, the goal for forage removal was approximately

50% of available biomass during a given grazing period. By leaving approximately half of the

available biomass, there was adequate plant material to perform photosynthesis, which

theoretically allowed for efficient regeneration of aboveground biomass while maintaining

carbohydrate reserves in the roots. The time, date and location of each move, as well as the size

of each paddock, were tracked using the PastureMap mobile application ("PastureMap Grazing

Management and Livestock Software," 2019).

Decisions on paddock size were made based on forage availability and soil conditions.

Biweekly assessments of forage yield were made and used to adjust future paddock sizes for the

number of cattle currently grazing. Cattle numbers fluctuated at certain times due to events such

as artificial insemination, embryo transfer, and calf vaccinations. For calculating paddock sizes,

first, the stocking density was estimated in kg of liveweight ha$^{-1}$ [Eqn. 1]. Available forage was

estimated based on kilograms of dry matter ha$^{-1}$ from hand clipped samples, 0.50 (or 50%) as the

desired forage utilization percentage, estimated daily intake of 2.6 to 3.0% of animal body

weight, and desired grazing duration before the next herd move, generally one day. Next,

paddock size (ha) [Eqn. 2] was calculated based on total kg of liveweight for the entire herd

divided by kg of liveweight ha$^{-1}$ determined from Eqn. 1.

$$\frac{\text{Available forage (kg DM ha}^{-1}) \text{ x } 0.50 \text{ (utilization goal)}}{\text{Daily intake (\% body weight) x Grazing duration (days)}} = \text{kg liveweight ha}^{-1} \text{ [Eqn. 1]}$$

$$\frac{\text{Total kg liveweight for herd}}{\text{kg liveweight ha}^{-1}} = \text{Paddock size in ha [Eqn. 2]}$$

*2.4 Soil Sampling and Processing*

Only the major soil series containing similar textures within each forage mixture were

sampled; areas of extreme variability were excluded (e.g., wet areas and small sections of



extraneous soil types).  The major soil series included Nunn clay loam, Kim loam, and Garrett
loam, which collectively comprised 74% of the total area under the center pivot. Soil samples
were collected for analysis before grazing in May 2017 and May 2018. Sampling locations were
randomly determined using ArcMap (Version 10.5.1, ArcMap GIS) and located using Avenza
Maps (Version 3.6, 96.17) when physically in the field. Five replicates were sampled within each
forage mixture with 30 soil cores per replicate collected within approximately a 3 m radius
surrounding each randomly determined sampling location. Soils were collected using a 3.2 cm
inner diameter soil probe, with samples split into 0 to 5 and 5-15 cm depths. Samples were
immediately placed in plastic bags, sealed, and placed in coolers. An extra core was obtained
from both depth increments for each replicate and placed in a metal can for gravimetric soil
moisture and bulk density determination.

Once returned to the laboratory, samples were stored in a refrigerator at 4ºC before

processing. Cores for moisture content and bulk density were weighed, then immediately dried at
105ºC for at least 24 hours, and then weighed again. The bulk soil samples were passed through
an 8-mm sieve, removing large pieces of organic material and rock. A representative sub-sample
of ~150 g of field-moist, 8-mm sieved soil was placed immediately in a plastic Ziplock bag,
labeled and stored at 4ºC for subsequent microbial biomass C analysis. Another sub-sample of
~150 g of 8-mm sieved soil was passed through a 2-mm sieve and then air-dried, while the
remaining 8-mm sieved soil was allowed to air-dry for subsequent analyses.
*2.5  Soil Health and Laboratory Soil Analyses*

The Soil Management Assessment Framework (SMAF) is an assessment tool that utilizes

11 soil indicators, in conjunction with soil taxonomy, climatic conditions, and management
practices, as a foundation for quantifying soil health (Andrews et al., 2004). Soil indicators



include bulk density and water stable aggregates (soil physical health indicators), soil organic
carbon, microbial biomass carbon, potentially mineralizable nitrogen, and beta-glucosidase
activity (soil biological health indicators), pH and electrical conductivity (EC; soil chemical
health indicators), and plant-available potassium and phosphorus (soil nutrient health indicators).
Soil texture, and more specifically clay content based on soil texture determination, influences
most indicators and is thus utilized in the background for soil health quantification in the SMAF.
Although most indicators are routinely quantified, others are not; quantification of all 11 soil
indicators are presented in the Supplemental Material. Once entered into the SMAF, individual
indicators are grouped into nutrient, chemical, physical, biological and overall soil health indices
(SHI). A soil's quantified properties, climatic conditions, how it is utilized, and management
practices performed are considered within the SMAF to create an output that reflects the specific
limitations and needs of the soil to function at its fullest potential. The SMAF has been
previously used to quantify soil health changes in native pasture, perennial vegetation systems,
and cropland converted to pasture (Veum, et al., 2015; Paudel et al., 2011), but has yet to be used
in irrigated MiG systems.
*2.6  Statistical Analysis*
An analysis of variance (ANOVA) using Kenward-Roger degrees of freedom with a test
significance level of $p \leq 0.05$ was performed on all indicator raw data, all indicator scores, and
for the physical, chemical, biological, nutrient, and overall SHI for the 0 to 5 and 5 to 15 cm
depths using RStudio Version 1.1.456 (R Core Team, 2017). The linear mixed effect model
utilized was created with the Lmer package in RStudio (Bates et al., 2015). Comparisons were
made for each soil indicator, indicator score, and all SHI between years, between depths, and the
interaction between year and depth.





**3 Results & Discussion**
*3.1 Soil Physical Indicators and Physical Soil Health*

A significant increase in bulk density was observed between 2017 and 2018 (Table 2).

This led to a significant decrease in the bulk density index score between years, from 0.80 to
0.37 and 0.59 to 0.34 in the 0-5 and 5-15 cm depths, respectively (Table 3). Soil surface bulk
density was likely lower post tillage due to soil mixing during ground preparation for planting
and would likely increase when cattle exert hoof pressure on the soil during grazing. The
minimum and maximum bulk densities measured in 2018 after the first grazing season were 0.77
and 1.89 g cm$^{-3}$, respectively. Previous studies have shown that when soils reach or exceed a
bulk density of 1.7 g cm$^{-3}$, root growth is impeded (Bruand and Gilkes, 2002). Although mean
bulk density levels did not reach 1.7 g cm$^{-3}$, future monitoring of this indicator will be important
from a soil health and forage productivity perspective.

Water stable aggregates (WSA) did not change significantly between pre- and post-

grazing (Table 2). However, a significant difference between depths for WSA indicator values
and index scores existed (Tables 3 and 4, respectively). Greater aggregation was present at the 5-
15 cm depth than the 0-5 cm depth. In 2018, mean WSA percentages were 45.7% at 0-5 cm and
59.6% at 5-15 cm. Factors that could have contributed to this difference include the lack of
tillage from 2017 to 2018, the addition of perennial grasses with fibrous root systems, and
microbial activity (e.g., increased MBC, discussed below) related to these management changes.
Soil aggregate formation relies heavily on microbial activity which is often greater in grazed,
improved pasture systems than in native or tilled systems, simply due to the level of production
present (Sparling, 1992; Warren et al., 1986). Less WSA in the 0-5 cm depth could be attributed
to the physical pressure of grazing. Warren et al. (1986) found that soil aggregate size was



negatively correlated to trampling rate. Although perennial vegetation and microbial activity can
aid in aggregation, pressure on the soil surface from animals' hooves during grazing may have an
adverse effect on aggregation.
Soil bulk density and WSA data both contribute to the physical SHI value in SMAF.
Changes in bulk density were the primary factor that caused a significant decrease in the physical
SHI between 2017 and 2018 (Table 4).
*3.2  Soil Biological Indicators and Biological Soil Health*
β-glucosidase activity significantly increased from 2017 to 2018 (Table 2). This was
likely the result of a land-use change from a tilled cropping system to a perennial pasture system.
Bandick and Dick (1999) concluded that BG activity responds to soil management practices due
to its role in the C cycle. The authors mentioned that an uninterrupted rhizosphere and greater
organic matter additions harbor greater levels of enzymatic activity; the concept of an
uninterrupted rhizosphere supports our findings. Martens et al. (2004) concluded that the upper
soil profile contained more β-glucosidase-type enzymes as long as management practices
contribute plant biomass and tillage is avoided. The increase in BG led to a significant increase
in the BG index score from 2017 to 2018 (Table 3). Bandick and Dick (1999) suggested that
enzymes in the soil may be early indicators of biological change when management practices are
altered. Future monitoring and analysis will be needed to observe if additional changes in soil
biological activity occur over time.
Microbial biomass C significantly increased from 2017 to 2018 (Table 2) resulting in a
greater MBC index in 2018 (Table 3). There was also a significant year by depth interaction due
to a greater increase at the 0-5 than 5-15 cm depth over time. Other studies have shown that
MBC is often greater in surface soils of systems without tillage due to surface residue acting as a



C (i.e., energy) source for  microbes (Doran, 1987). Converting from a tilled, low residue system
to a perennial, grazed system likely provided an influx of C material responsible for the increase
in MBC. In addition, the grazing strategy employed in this study aimed to utilize only 50% of the
forage biomass present during a grazing period. According to multiple studies, partial plant
defoliation has been found to increase soluble exudates from plant roots (Bardgett et al., 1998;
Holland et al., 1996). This rhizodeposition, in turn, would be expected to stimulate soil microbial
activity. Synthetic N was not added to this system once converted to a perennial pasture.
Reduced soil N has been found to stimulate the release of organic root exudates in grasses as
well as foster a greater microbial community (Bardgett et al., 1998; Hodge et al., 1996). Studies
have shown that cool-season, managed grasses, like those present in the current study, tend to
exude quickly decomposable C substrates which can stimulate microbial activity (Grayston et al.,
1998). Easily decomposable C substrates were also likely present in this grazing system due to
cattle manure inputs and the fast-growing nature of modern cool-season grass varieties (Dawson
et al., 2000).

Potentially mineralizable nitrogen was significantly greater in 2018 than in 2017 (Table

2), causing a significant increase in the PMN index score from 2017 to 2018 over both soil
depths (Table 3). Precipitation (or irrigation in the current study) has been positively correlated
with PMN soil concentrations (Doran, 1987), with PMN serving as an indicator of a microbial
population's capacity to mineralize nitrogen from organic to plant-available forms. Thus,
managed irrigation could provide an advantage to MiG systems in terms of how quickly manure
N is mineralized; a further advantage would be that perennial pasture systems are not tilled.
Doran (1987) found that in a no-till system, soil microbial biomass and PMN distributions were
similar, with both being greatest in the top 7.5 cm of soil. In long-term grazing systems, manure



and plant litter decomposition are the main fertility sources, yet are only found on the soil
surface and are not incorporated. Thus, having proportionately more MBC and PMN in the top
few cm of soil is advantageous for plant material degradation and nutrient cycling in MiG
systems.
Soil organic C remained unchanged from 2017 to 2018, resulting in no significant change
to the SOC index value (Tables 2 and 3). It should be noted that changes in BG and MBC have
been detectable earlier than changes in SOC because of the rapid turnover rate, with BG and
MBC being early indicators of long-term soil C accumulation (Sparling, 1992; Turner et al.,
2002). Given time, we would expect the system in the current study to significantly gain SOC,
which along with increased WSA under pasture settings, would lead to physical soil
improvements (Martens et al., 2004). Continued monitoring will be necessary to track possible
SOC changes over time and correlations with other indicators in this system.
The changes that occurred in three out of the four biological indicators caused an increase
in the biological soil health index score from 2017 to 2018 (Table 4). The land-use change from
a tilled, cropping system to a no-till perennial system has likely imparted positive changes on soil
biological activity and, thus, the biological soil health index. Veum et al. (2015) utilized the
SMAF to assess soil health for different annual and perennial cropping systems. They concluded
that biological and physical SHI categories were the most sensitive to changes in management.
Paudel et al. (2011) found that grazed pasture systems had greater β-glucosidase activity and soil
organic C. They concluded that because there is minimum disturbance, more organic matter can
accumulate resulting in ecological benefits to the system. Again, given time, we would expect
the current pasture system to gain soil organic C.
*3.3 Soil Chemical Indicators and Chemical Soil Health*



There was no significant change in pH or the pH index value from 2017 to 2018 (Tables
2 and 3). Due to percent calcium carbonate (7 to 15%) and CEC (10 to 28 cmolc kg$^{-1}$) (CA Soil
Resource Lab, 2008), in addition to clay content (28 to 49%; as determined for the SMAF), soils
at this site likely have a high buffering capacity that resists change in pH. This could mean that
pH, even over the future long-term of this grazing project, may not significantly change.
Electrical conductivity significantly decreased from 2017 to 2018, resulting in a
significant increase in the EC indicator score (Tables 2 and 3). Electrical conductivity is an
important indicator of soil health in agroecosystems and can be impacted by management
changes in relatively short periods of time. Inherent soil properties, such as texture and parent
material, as well as management practices like irrigation, fertilization, and land-use, all influence
EC (USDA-NRCS, 2014).The reduction in fertilizer inputs (only applied at time of seeding)
since the land-use change, combined with irrigation, could explain the EC decrease simply due to
flushing of fertilizer-borne salts below the 15 cm soil depth. The EC reduction led to a
significant increase in the chemical soil health index between depths (Table 4).
*3.4  Soil Nutrient Indicators and Nutrient Soil Health*
Extractable K concentrations significantly increased from 2017 to 2018, the 0-5 cm depth
contained greater extractable K than the 5-15 cm depth, and a significant year by depth
interaction existed (Table 2). The extractable K index values were significantly different between
years and between depths with a higher indicator score for the 0-5 cm depth (Table 3); a
significant interaction between year and depth was also present due to the greater increase in K
in the 0-5 cm depth compared to the 5-15 cm depth in 2018. The increase in soil K concentration
in the 0-5 cm depth was likely due to urine deposition from cattle grazing. Approximately 73%
of K consumed by cattle is excreted in urine and is often 100% bioavailable (Eghball et al.,



2002). Early et al. (1998) studied the fate of K in simulated urine patches under irrigated grazing
of dairy cattle using lysimeters, finding that ~20% of K that was applied remained within the top
0-5 cm of soil.

Olsen-extractable P significantly decreased from 2017 to 2018 (Table 2) leading to a

decrease in the extractable P index value between years (Table 3).  Approximately 96% of P
intake by cattle is excreted in manure, 70% of which is primarily in inorganic forms (Eghball et
al., 2002). Because of this, extractable P concentrations were expected to increase due to cattle
manure deposition, yet this was not the case. The decrease in soil extractable P concentration led
to a significant reduction in the nutrient SHI (Table 4).  The decrease could have been the result
of several factors. First, the equivalent of 54 kg $P_2O_5$ ha$^{-1}$ was applied in August 2016 just prior
to seeding. There would have been minimal uptake of the applied P at time of the first soil
sampling in May 2017. By the time the second set of soil samples were taken in May 2018, a full
growing season had elapsed which would account for a significant amount of plant P uptake and
removal, especially since initial growth was harvested for hay in June 2017. Second, manure
deposits average 0.12 m$^2$ in area compared to urine deposits which average 0.36 m$^2$ (Wilkinson
and Lowery, 1973). Therefore, the likelihood of seeing an impact on K concentrations was three
times greater than the potential impact of manure deposition on P concentrations in the soil.
*3.5  Combined Effects on Physical, Biological, Chemical, and Nutrient Soil Health on Overall*
*Soil Health*

Physical and nutrient SHI's both decreased, while the biological SHI increased between

years.  These changes essentially negated each other, leading to no significant change in the
overall soil health index from 2017 to 2018 (Table 4). Previous observations of positive and
negative changes in component SHI values, leading to no improvement in overall soil health,





have been noted in other ecosystems (Ippolito et al., 2019). Although a significant overall soil
health change was not measured at the present time, as soil parameters shift over time due to the
continued influence of grazing and having the system in perennial forages, they may cause
future, significant shifts in this indicator. Thus, additional future monitoring is required.



**4 Conclusions**

Soil physical, biological, and nutrient SHI values responded significantly to management changes from a tilled, irrigated cropping system to a no-till, irrigated perennial MiG system. Positive soil health effects were observed in the biological SHI, in particular, increases in MBC and BG enzymatic activity, both of which could be early indicators of future C sequestration. These findings support our hypothesis that the biological SHI would increase under MiG. In the future, the addition of microbial level physiological profiling (e.g., proportion of bacteria:fungi) would increase our understanding of the implications of management-intensive grazing on soil biological health. Soil organic C remained relatively unchanged but will be an important indicator to monitor into the future, especially with regards to its link with MBC and PMN.

Negative impacts occurred to the physical SHI, driven primarily by increasing bulk density. This finding supports our hypothesis that the physical SHI would decrease under MiG. Furthermore, this result was likely caused by initial hoof compression of the soil surface during grazing. Bulk density is an indicator that should be monitored closely in the future due to its potential impacts on hydrology and root health. The nutrient SHI value declined due to the observed reduction in extractable soil P, which did not support our hypothesis that this SHI would increase under MiG. Cattle urine inputs likely contributed to a significant increase in available K, which would also have been expected for P due to its high concentration in cattle manure (Eghball et al., 2002). However, the opposite was observed, ultimately reducing mean P concentrations from 2017 to 2018, and negatively affecting the nutrient SHI. Although these are only initial observations in the early stages of conversion, irrigated MiG systems appear to have the potential for success with regards to supporting grazing while promoting soil health for environmental and economic sustainability.



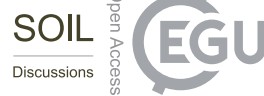

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

Methods of Soil Analysis, Part 3 –Chemical Methods. Soil Science Society of America,

Madison, WI, pp. 417-435.

Shand, C.A., Macklon, A.E.S., Edwards, A.C., Smith, S., 2006. Inorganic and organic P in soil

solutions from three upland soils. Plant Soil 160, 161–170.

Sherrod, L.A., Dunn, G., Peterson, G.A., Kolberg, R.L., 2002. Inorganic C analysis by modified

pressure-calcimeter method. Soil Sci. Soc. Am. J. 66, 299.

Shewmaker, G.E., Bohle, M.G., 2010. Pasture and Grazing Management in the Northwest.

Pacific Northwest Ext. Pub. PNW 214. Univ Idaho Ext, Moscow, ID. 208 pp.

Slavich, P.G., Petterson, G.H., 1993. Estimating the electrical conductivity of saturated paste

extracts from 1:5 soil, water suspensions and texture. Aust. J. Soil Res. 31, 73–81.

Sparling, G.P., 1992. Ratio of microbial biomass C to soil organic C as a sensitive indicator of

changes in soil organic matter. Aust. J. Soil Res. 30, 195–207.

Stout, W.L., Fales, S.L., Muller, L.D., Schnabel, R.R., Elwinger, G.F., Weaver, S.R., 2000.

Assessing the effect of management intensive grazing on water quality in the northeast U.S.

527          J. Soil Water Conserv. 55, 238-243.

Thomas, G.W., 1996. Soil pH and soil acidity. In: Sparks, D.L. (Ed.), Methods of Soil Analysis,

Part 3 –Chemical Methods. Soil Science Society of America, Madison, WI, pp. 475-490.

Tisdall, J.M., Oades, J.M., 1982. Organic matter and water-stable aggregates in soils. J. Soil Sci.

33, 141–163.

Turner, B.L., Hopkins, D.W., Haygarth, P.M., Ostle, N., 2002. β-glucosidase activity in pasture

533 soils. Appl. Soil Ecol. 20, 157–162.

534 USDA-NRCS, 2014. Soil electrical conductivity. United States Dept. Agri. Natural Res. Cons.

535 Ser. Soil Health Guide for Educators.

536 https://www.nrcs.usda.gov/Internet/FSE_DOCUMENTS/nrcs142p2_052803.pdf (accessed

537 1 October 2019).

538 Veum, K.S., Kremer, R.J., Sudduth, K.A., Kitchen, N.R., Lerch, R.N., Baffaut, C., Stott, D.E.,

539 Karlen, D.L., Sadler, E.J., 2015. Conservation effects on soil health indicators in the

540 Missouri Salt River Basin. J. Soil Water Conserv. 70, 232–246.

541 Warren, S.D., Nevill, M.B., Blackburn, W.H., Garza, N.E., 1986. Soil response to trampling

542 under intensive rotation grazing. Soil Sci. Soc. Am. J. 50, 1336-1341.

543 Wilkinson, S.R., Lowrey, R.W., 1973. Cycling of mineral nutrients in pasture ecosystems. p.

544 247–315. In: G.W. Butler and R.W. Bailey (ed.), Chemistry and Biochemistry of Herbage.

545 Vol. 2. Academic Press, New York.



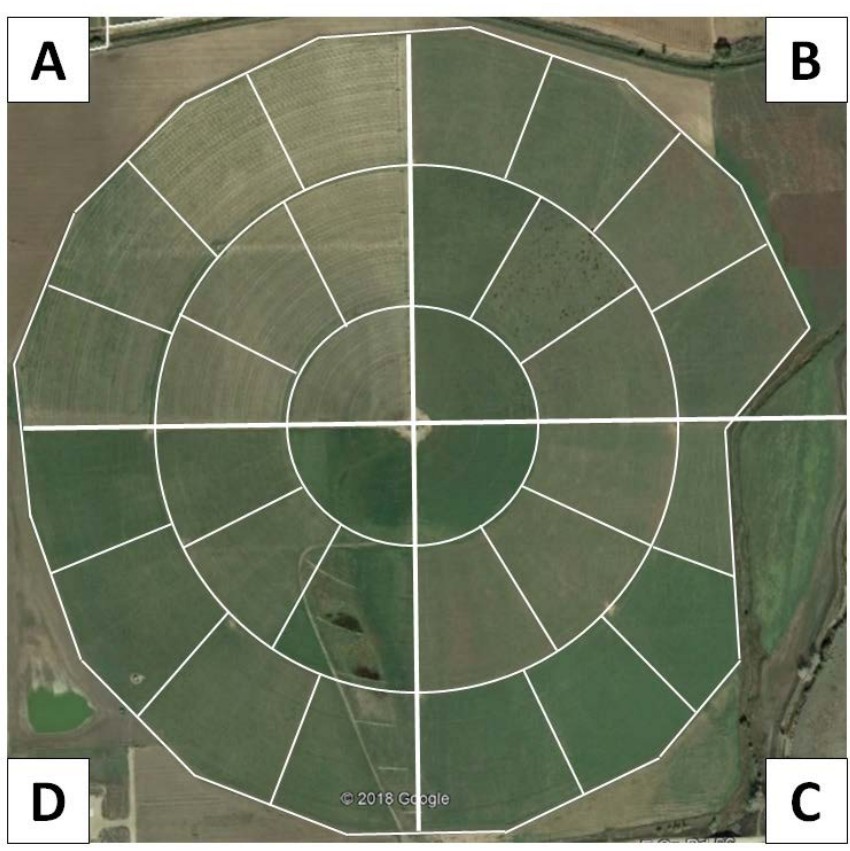

Figure 1.  Diagram of eight pre-determined paddocks within each cool season mixture (A, B, C,
D; ~20 ha each). White lines roughly represent electric fence locations.



Table 1. Forage species planted in each ~20-ha portion (A, B, C, or D) of the 82 ha project area.

| Forage Mixture | Species |
|---|---|
| A - Grass/Legume Mix | Meadow brome (*Bromus biebersteinii Roem. & Shult.*), Orchardgrass (*Dactylis glomerata L.*), Creeping meadow foxtail (*Alopecurus arundinaceus* Poir.*), Birdsfoot trefoil (*Lotus corniculatus L.*), Strawberry clover (*Trifolium fragiferum L.*), White clover (*Trifolium repens L.*) |
| B - Complex Grass Mix | Meadow brome, Orchardgrass, Creeping meadow foxtail, Tall fescue (*Festuca arundinacea* Shreb.), Festulolium (*xFestulolium*), Smooth brome (*Bromis inermis L.*) |
| C - Simple Grass Mix | Meadow brome, Orchardgrass, Creeping meadow foxtail |
| D - Complex Grass/Legume Mix | Meadow brome, Orchardgrass, Tall fescue, Perennial ryegrass (*Lolium perenne*), Meadow fescue (*Festuca pratensis*), Festulolium (*xFestulolium*), Red clover (*Trifolium pretense L.*), Alsike clover (*Trifolium hybridum L.*), White clover, Birdsfoot trefoil |





Table 2. Mean Soil Management Assessment Framework individual soil indicator values in years and
depths, and analysis of variance (ANOVA) between years, depths, and the year by depth interaction,
within a management-intensive, irrigated grazing system.

| Soil Indicator | 2017 | 2018 | 2017 | 2018 | ANOVA (between years) | ANOVA (between depths) | ANOVA (year by depth) |
|---|---|---|---|---|---|---|---|
| | 0 to 5 cm | | 5 to 15 cm | | | | |
| $\rho_b$ (g cm$^{-3}$)$^\dagger$ | 1.15 | 1.52 | 2.91 | 2.44 | **ꞯ | NS | NS |
| WSA (g kg$^{-1}$) | 43.1 | 45.7 | 57.2 | 59.6 | NS | ** | NS |
| BG (mg pnp kg$^{-1}$ soil h$^{-1}$) | 72.1 | 88.2 | 69.8 | 76.0 | ** | NS | NS |
| PMN (mg kg$^{-1}$) | 15.6 | 18.3 | 12.8 | 16.8 | ** | NS | NS |
| SOC (%) | 1.55 | 1.40 | 1.40 | 1.45 | NS | NS | NS |
| MBC (mg g$^{-1}$) | 118 | 316 | 132 | 245 | ** | NS | * |
| pH | 7.98 | 8.11 | 7.93 | 8.03 | NS | NS | NS |
| EC (dS m$^{-1}$) | 2.18 | 1.34 | 2.91 | 2.44 | ** | NS | NS |
| K (mg kg$^{-1}$) | 204 | 415 | 195 | 253 | ** | ** | ** |
| P (mg kg$^{-1}$) | 16.1 | 13.3 | 11.7 | 11.1 | ** | NS | NS |

$^\dagger$ $\rho_b$ = bulk density, WSA = water-stable aggregates, BG = β-glucosidase activity (pnp = $p$-nitrophenol),
PMN = potentially mineralizable N, SOC = soil organic C, MBC = microbial biomass C; EC = electrical
conductivity, K = extractable K, and P = extractable phosphorus.
ꞯ NS = non-significant, * = $p < 0.05$, and ** = $p < 0.01$.





Table 3. Soil Management Assessment Framework individual soil indicator index scores (0.00 to 1.00;
greater is better) in years and depths, and analysis of variance (ANOVA) between years, depths, and the
year by depth interaction, within a management-intensive, irrigated grazing system.

| Soil Indicator | 2017 | 2018 | 2017 | 2018 | ANOVA (between years) | ANOVA (between depths) | ANOVA (year by depth) |
|---|---|---|---|---|---|---|---|
| | 0 to 5 cm | | 5 to 15 cm | | | | |
| $\rho_b$ (g cm$^{-3}$)$^\dagger$ | 0.81 | 0.37 | 0.60 | 0.35 | **⌐ | * | NS |
| WSA (g kg$^{-1}$) | 0.78 | 0.73 | 0.91 | 0.88 | NS | ** | NS |
| BG (mg pnp kg$^{-1}$ soil h$^{-1}$) | 0.05 | 0.06 | 0.05 | 0.06 | ** | NS | NS |
| PMN (mg kg$^{-1}$) | 0.82 | 0.97 | 0.68 | 0.90 | ** | NS | NS |
| SOC (%) | 0.19 | 0.16 | 0.17 | 0.16 | NS | NS | NS |
| MBC (mg g$^{-1}$) | 0.10 | 0.54 | 0.12 | 0.36 | ** | NS | * |
| pH | 0.02 | 0.01 | 0.02 | 0.01 | NS | NS | NS |
| EC (dS m$^{-1}$) | 0.54 | 0.82 | 0.30 | 0.46 | ** | NS | NS |
| K (mg kg$^{-1}$) | 0.96 | 1.00 | 0.94 | 0.94 | ** | ** | ** |
| P (mg kg$^{-1}$) | 0.94 | 0.60 | 0.75 | 0.38 | ** | NS | NS |

$^\dagger$ $\rho_b$ = bulk density, WSA = water-stable aggregates, BG = β-glucosidase activity (pnp = $p$-nitrophenol),
PMN = potentially mineralizable N, SOC = soil organic C, MBC = microbial biomass C; EC = electrical
conductivity, K = extractable K, and P = extractable phosphorus.
⌐ NS = non-significant, * = $p < 0.05$, and ** = $p < 0.01$.



Table 4. Soil Management Assessment Framework physical, chemical, biological, nutrient, and overall
soil health index scores (0.00 to 1.00; greater is better) in years and depths, and analysis of variance
(ANOVA) between years, depths, and the year by depth interaction, within a management-intensive,
irrigated grazing system.

| Soil Health Index | 2017 | 2018 | 2017 | 2018 | ANOVA (between years) | ANOVA (between depths) | ANOVA (year by depth) |
|---|---|---|---|---|---|---|---|
| | 0 to 5 cm | | 5 to 15 cm | | | | |
| Physical | 0.79 | 0.55 | 0.75 | 0.62 | **⊦ | NS | NS |
| Biological | 0.29 | 0.43 | 0.26 | 0.37 | ** | NS | NS |
| Chemical | 0.28 | 0.42 | 0.16 | 0.24 | NS | * | NS |
| Nutrient | 0.95 | 0.80 | 0.85 | 0.66 | ** | ** | NS |
| Overall | 0.52 | 0.53 | 0.45 | 0.45 | NS | NS | NS |

⊦ NS = non-significant, * = p < 0.05, and ** = p < 0.01.