# Peer review of "Management-intensive Grazing Affects Soil Health"

_SOIL, 2019_

## Referee Comment (RC1) · Anonymous Referee #1 · 4 Mar 2020

The study has addressed potential 'Management-intensive grazing' (MiG) effects on soil health represented by 11 soil variables. Results show significant positive changes in soil biological properties after the land use change (from cropland-to-grassland) and negative effects on soil bulk density and soil P availability. The authors conclude that MiG could have positive benefits for soil health and environmental and economic sustainability. Although the results of this study are interesting (increases in microbial biomass and extra-cellular enzyme activity, decreases in bulk density), it is very difficult to state that MiG is actually responsible for these changes. This is because: - The study measured these variables only 1 and 2 years after soils had been under cropping for at least 10 years. Thus changes in soil variables could be simply due to a change in land use from cropland-to-grassland and/or to water irrigation (and thus to

increased soil moisture) and not necessarily to MiG. - The impact of MiG was tested only between August 2017 (when cows were firstly introduced to paddocks) to May 2018 (when soil samples were taken for the last time). Thus comparing changes in soil properties between 2017 and 2018 when grazing was applied only between August and October 2017 leaves lots of uncertainty about potential grazing effects on soil health. Most of these changes could be simply related to the land use change and the establishment of a grassland ecosystem. To address MiG effects on soil health data from at least few years should be collected and MiG should be compared to a different permanent grassland system perhaps not irrigated. - The fact that the grass swards did not establish well due failure of legume growth 2017 and 'hostile' climatic conditions between 2017 and 2018 adds more uncertainty to net effects of grazing on soil health. I would collect data from more years to provide evidence that MiG is benefiting soil health, which could well be the case.

Specific comments Lines32-34: I would rephrase the last sentence, which at the moment seems to consider only MiG positive effects on soil biology but not negative MiG effects on soil P availability and soil structure (increased compaction). Given these contrasting results one could argue why the authors conclude that MiG is 'promoting soil health for environmental and economic sustainability'. Line 40: after 'benefits' needs literature reference(s). Lines 46-49: This sentence needs rephrasing to introduce the definition/concept of 'soil health' supported by literature references. Lines 56-60: These lines would need to be rewritten to explain better the role of extracellular enzymes in soils. In particular, microbes produce extra cellular enzymes to acquire C or nutrients from SOM and as a consequence affect C, N dynamics. Line 93: Needs to explain/summarize why grazing has positive effects on soil 'health'. Lines 112: I am not sure that hypothesis two is well supported. For example, microbial biomass C and enzyme activities can be stimulated by animal waste more than lack of tillage? Need to better support this hypothesis in the Introduction. Lines 136-139: Perhaps few lines explaining the rationale behind species assemblages (e.g. simple vs complex) would be useful. Line 174: I think this is August 2017 and not 2018 Lines 210: A significant

problem associated with the robustness of the dataset assembled under this project is that potential grazing effects on soil health have been addressed only after one grazing season between August 2017 (when cows were firstly introduced to paddocks) to May 2018 (when soil samples were taken for the second time). This is quite a short period of time to address potential grazing effects on multiple soil parameters given that cows grazed only between August and October 2017 on swards which had not established well (legumes were almost absent from swards), the soils were cropped until 2016 and climatic conditions were quite variable and 'hostile' at 1500 m asl. If more soils had been collected in 2019 (after at least two grazing seasons) this would have been better and provided more data to compare... Lines 236-245: It is not clear how the SMAF works, how field measurements are 'transformed' in indexes through SMAF. More details need to be given here. Lines 246-253: Statistical analyses are not properly described, clear description of independent and depend variables is not given. 'Years' is the variable indicator of potential grazing effects on soil parameters? Lines 284-285: The fact that BG activity increases it could be due (as the authors suggest) to land use change from cropland to pasture. This could occur however following any cropland-to-grassland land use change and not necessarily only to MiG on irrigated grasslands. Also there is a problem with 'perennial' pasture because the grassland in this study is only 1 year old. Lines 296-300: Increases in microbial biomass, again, could be related to the land use change and not necessarily to MiG. To test for the effects of MiG on multiple soil parameters, data should be collected at least for few years and also compared to permanent grasslands, which are not irrigated for example. Other variables such as soil N mineralization could have increased because of greater soil moisture due to irrigation and not necessarily to MiG. It is not surprising that soil C has not changed because of the short period of time considered (2017-2018) (line 372) It is actually surprising that soil P availability decreased (and not increased) under the MiG system with more cattle dung and urine being returned to soils. I think this could be due to the fact that most P has been retained by soils and partly perhaps because more P was uptaken by plants. This however shows the difficulty to interpret these results after

only a short grazing season had occurred on these newly established grasslands.

---

## Referee Comment (RC2) · Anonymous Referee #2 · 13 Mar 2020

The manuscript is dealing with the question if land-use-change from crop to pasture would affect the soil properties. The manuscript is general well written however, there are three major concerns I have with the presented results: i) very short-term effects (only 1-2 year after LUC) are discussed, ii) no control treatment (e.g. no grazing) iii) no randomized established replicates. Moreover, there was a lot of effort spent to introduce the different forage mixtures but results of this factor are either discussed very extensively or are not present. Its clear that the main effect on soil function, in this initial phase, is the land-use change to grassland rather than due to the effect of grazing animals. However, this statement cannot be confirmed accurately as there were no, as far as I understood, non-grazed paddocks present in this trial. I am wondering if authors could make a clearer statement about the grazing yields and the differences

between the mixtures in the results & discussion chapter. I guess due to the long-arable crop phase as pre-management and the extensive nitrogen application there are low biomass yields in comparison to other irrigated and well fertilized pastures leading into a reduction of plant residues, which in turn reduces the assumed carbon sequestration rate. This fact was maybe additionally triggered by the poor forage legume establishment even though high soil pH-values should allow favorable growing conditions. In addition, there are only a few information about the dairy herd available (e.g. breed) particularly with regards to the feeding strategy (e.g. supplements) and consequently about the potential nutrient excretion of grazing cattle, which make results of chemical soil properties hard to explain. Line 126: Mean of what? Monthly I guess Line 251: that you used RStudio is not relevant in this context. Line 251: as this is not a classical experimental design I am wonder if repeated measurements should be considered in your model. Line 262: seems to be very heterogenic. Give SD or SE.# Line 328: You explained that bulk densities increased. Even when this was observed with a high variation, soil carbon stocks in turn should be higher, if C content remained constant?! Table2 and 3: Table 2 is for me personally more helpful as the used index. Actually, I have my doubts about the benefits of this used index to understand the presented results.

---

## Author Comment (AC1) · 30 Mar 2020

The study has addressed potential 'Management-intensive grazing' (MiG) effects on soil health represented by 11 soil variables. Results show significant positive changes in soil biological properties after the land use change (from cropland-to-grassland) and negative effects on soil bulk density and soil P availability. The authors conclude that MiG could have positive benefits for soil health and environmental and economic sustainability. Although the results of this study are interesting (increases in microbial biomass and extra-cellular enzyme activity, decreases in bulk density), it is very difficult to state that MiG is actually responsible for these changes. This is because: -

The study measured these variables only 1 and 2 years after soils had been under cropping for at least 10 years. Thus changes in soil variables could be simply due to a change in land use from cropland-to-grassland and/or to water irrigation (and thus to increased soil moisture) and not necessarily to MiG. - The impact of MiG was tested only between August 2017 (when cows were firstly introduced to paddocks) to May 2018 (when soil samples were taken for the last time). Thus comparing changes in soil properties between 2017 and 2018 when grazing was applied only between August and October 2017 leaves lots of uncertainty about potential grazing effects on soil health. Most of these changes could be simply related to the land use change and the establishment of a grassland ecosystem. To address MiG effects on soil health data from at least few years should be collected and MiG should be compared to a different permanent grassland system perhaps not irrigated. The fact that the grass swards did not establish well due failure of legume growth 2017 and 'hostile' climatic conditions between 2017 and 2018 adds more uncertainty to net effects of grazing on soil health. I would collect data from more years to provide evidence that MiG is benefiting soil health, which could well be the case.

Authors Reply: First, we appreciate this reviewer's comments with respect to the issues with such a short study. We agree that collecting more data would be best. Unfortunately we only received two years of funding for this project and don't have the financial support to continue at this moment in time. We are attempting to find additional funding to continue this work. We would state that the comment of collecting more data likely could be applied to the majority of published field studies, since most field studies are simply not long-term. Our study also falls within that realm.

We also agree that the changes may have been due to converting the system from a cropping to a pasture system. BUT, this is why we were careful with our wording in the manuscript. The reviewer even pointed this out above that: "MiG could have positive benefits for soil health and environmental and economic sustainability". Again, our careful word selection so as to not be exact with our thought process as to why

these processes were occurring.

Lastly, we can tell you that others have found similar, short-term results with MiG in different ecosystems. If you visit this article: https://onpasture.com/2016/06/20/does-management-intensive-grazing-grow-more-better-quality-forage/ that describes MiG research from University of Madison (Wisconsin, USA), the researchers found that MiG soils lost less C than other treatments, at least in one year of their two year study. They also found no change in soil microbial activity over their short-term study when comparing treatments to MiG. Furthermore, the authors found greater net N mineralization with MiG over ungrazed systems, in their short-term (i.e., two-year) study. It appears that there is an associated, peer-reviewed article from these researchers, found here: https://www.onpasture.com/wp-content/uploads/2016/01/Management_IntensiveRotationalGrazing.pdf We, unfortunately, did not cite this work in the original manuscript. If possible, we will work this into the revised manuscript in key locations, and will point out those locations using track changes.

Specific comments

Lines32-34: I would rephrase the last sentence, which at the moment seems to consider only MiG positive effects on soil biology but not negative MiG effects on soil P availability and soil structure (increased compaction). Given these contrasting results one could argue why the authors conclude that MiG is 'promoting soil health for environmental and economic sustainability'.

Authors Reply: Agreed and rewritten as: If managed correctly, compaction and soil P issues could be avoided, with MiG systems having potential success in supporting grazing while promoting soil health for environmental and economic sustainability.

Line 40: after 'benefits' needs literature reference(s).

Authors Reply: We added a citation.

Lines 46-49: This sentence needs rephrasing to introduce the definition/concept of 'soil health' supported by literature references.

Authors Reply: Rewording and added references, including one of our own to show that we have been working on soil health as well.

Lines 56-60: These lines would need to be rewritten to explain better the role of extracellular enzymes in soils. In particular, microbes produce extra cellular enzymes to acquire C or nutrients from SOM and as a consequence affect C, N dynamics.

Authors Reply: By rewriting these lines, it drives the discussion away from the point we are trying to make, which is that beta-glucosidase is used as a means of (partially) quantifying soil microbial health in the framework we utilized for this study. We did not make a change to this sentence.

Line 93: Needs to explain/summarize why grazing has positive effects on soil 'health'.

Authors Reply: This statement was indirectly referring to the aforementioned information above. We clarified the sentence.

Lines 112: I am not sure that hypothesis two is well supported. For example, microbial biomass C and enzyme activities can be stimulated by animal waste more than lack of tillage? Need to better support this hypothesis in the Introduction.

Authors Reply: We actually feel that we did a nice, concise job of writing the supporting information in the introduction. We focused an entire paragraph on this hypothesis. In the original manuscript, see lines 50 to 65. In the revised manuscript, see lines 56 to 74.

Lines 136-139: Perhaps few lines explaining the rationale behind species assemblages (e.g. simple vs complex) would be useful.

Authors Reply: Good suggestion. Information was added here.

Line 174: I think this is August 2017 and not 2018

Authors Reply: In the original manuscript, Line 174 (revised manuscript, line 185), the verbiage is correct. On August 18 (2017). However, we added "2017" after August 18 to be clearer.

Lines 210: A significant paper problem associated with the robustness of the dataset assembled under this project is that potential grazing effects on soil health have been addressed only after one grazing season between August 2017 (when cows were firstly introduced to paddocks) to May 2018 (when soil samples were taken for the second time). This is quite a short period of time to address potential grazing effects on multiple soil parameters given that cows grazed only between August and October 2017 on swards which had not established well (legumes were almost absent from swards), the soils were cropped until 2016 and climatic conditions were quite variable and 'hostile' at 1500 m asl. If more soils had been collected in 2019 (after at least two grazing seasons) this would have been better and provided more data to compare. . .

Authors Reply: See our reply to this author's first comment, above.

Lines 236-245: It is not clear how the SMAF works, how field measurements are 'transformed' in indexes through SMAF. More details need to be given here.

Authors Reply: Information has been added, and the reader is directed to Andrews et al. (2004) for additional details.

Lines 246-253: Statistical analyses are not properly described, clear description of independent and depend variables is not given. 'Years' is the variable indicator of potential grazing effects on soil parameters?

Authors Reply: The statistical description seems clear to us. We used a linear mixed effects model as suggested by our university's statistician, with comparisons made between the various environmental factors and indictors as listed on: a) in the original manuscript, lines 251-253; or b) in the revised manuscript, lines 264-267. No change was performed.

Lines 284-285: The fact that BG activity increases it could be due (as the authors suggest) to land use change from cropland to pasture. This could occur however following any cropland-to-grassland land use change and not necessarily only to MiG on irrigated grasslands. Also there is a problem with 'perennial' pasture because the grassland in this study is only 1 year old.

Authors Reply: Again, see our reply to the reviewer's first comment above. We do, however, back up this claim that the observed change may have been due to land use change by citing other works within this paragraph.

We clarified that the perennial pasture was only 1 year old in the text in these lines.

Lines 296-300: Increases in microbial biomass, again, could be related to the land use change and not necessarily to MiG. To test for the effects of MiG on multiple soil parameters, data should be collected at least for few years and also compared to permanent grasslands, which are not irrigated for example. Other variables such as soil N mineralization could have increased because of greater soil moisture due to irrigation and not necessarily to MiG. It is not surprising that soil C has not changed because of the short period of time considered (2017-2018) (line 372) It is actually surprising that soil P availability decreased (and not increased) under the MiG system with more cattle dung and urine being returned to soils. I think this could be due to the fact that most P has been retained by soils and partly perhaps because more P was uptaken by plants. This however shows the difficulty to interpret these results after only a short grazing season had occurred on these newly established grasslands.

Authors Reply: We agree with the reviewer that this observation could have been due to the recent land use change. Thus, we added a sentence that states that this could have been due to land use change (see lines 312-313 in the revised manuscript). This sentence still leads the reader into the remainder of the discussion that we had originally written.

The reviewer is still concerned about this being a short-term study based on the above

comment, and we again refer the editor to our reply to the reviewer's first comment above.

Please also note the supplement to this comment:
https://www.soil-discuss.net/soil-2019-91/soil-2019-91-AC1-supplement.pdf

---

## Author Comment (AC2) · 30 Mar 2020

The manuscript is dealing with the question if land-use-change from crop to pasture would affect the soil properties. The manuscript is general well written however, there are three major concerns I have with the presented results: i) very short-term effects (only 1-2 year after LUC) are discussed, Authors Reply: We addressed this comment as reviewer #1's first comment dealt with this issue as well. ii) no control treatment (e.g. no grazing)

Authors Reply: Good point. We did not include a control, but would a rancher ever include a control? No they would not. We tried to set this project up as practical as

possible because 1) it was a short-term funded research project, and 2) the forage was used to feed the Colorado State University beef cattle on the research site. Every parcel of ground was asked to be used.

iii) no randomized established replicates.

Authors Reply: We really couldn't set up this set as suggested by the reviewer, because there was soil variability across the site. Combining soil variability with fencing to separate paddocks for MiG was nearly impossible given the site configuration/dynamics. Thus why we sampled only the major, similar soil series on-site within each of the four forages under MiG. Basically, this site was impossible to set up as a traditional, say RCB with four replicates due to extreme on-site variability. We chose to soil sample only the major soil series within each of the four forage treatments in order to reduce that variability.

Moreover, there was a lot of effort spent to introduce the different forage mixtures but results of this factor are either discussed very extensively or are not present.

Authors Reply: The forage mixture comparison results are currently being written up for a separate manuscript.

Its clear that the main effect on soil function, in this initial phase, is the land-use change to grassland rather than due to the effect of grazing animals. However, this statement cannot be confirmed accurately as there were no, as far as I understood, non-grazed paddocks present in this trial.

Authors Reply: We do not entirely agree with this statement. See our reply to reviewer #1's first comment, where we cite work from the University of Wisconsin and their short term MiG soils research.

I am wondering if authors could make a clearer statement about the grazing yields and the differences between the mixtures in the results & discussion chapter. I guess due to the long-arable crop phase as pre-management and the extensive nitrogen application

there are low biomass yields in comparison to other irrigated and well fertilized pastures leading into a reduction of plant residues, which in turn reduces the assumed carbon sequestration rate. This fact was maybe additionally triggered by the poor forage legume establishment even though high soil pH-values should allow favorable growing conditions.

Authors Reply: The forage mixture comparison results are currently being written up for a separate manuscript.

In addition, there are only a few information about the dairy herd available (e.g. breed) particularly with regards to the feeding strategy (e.g. supplements) and consequently about the potential nutrient excretion of grazing cattle, which make results of chemical soil properties hard to explain.

Authors Reply: The breeds (Angus and Hereford) were presented in the original manuscript on line 175 (now line 186 in the revised manuscript). Other than that, the way the herd was managed and their feeding strategy throughout the in-field seasons, within the MiG system was outlined within the M+M section.

Specific comments

Line 126: Mean of what? Monthly I guess

Authors Reply: Correct, monthly. We made the change to the manuscript for clarity.

Line 251: that you used RStudio is not relevant in this context.

Authors Reply: We did not make a change to the manuscript, as we (and others) typically state the statistical software packages used in our statistics section of our manuscripts. Our graduate student performed this work, but when I run stats I use SAS version 9.4 and always cite it in this type of section.

Line 251: as this is not a classical experimental design I am wonder if repeated measurements should be considered in your model.

Authors Reply: We consulted our on-campus statistician for guidance and the designed we used was suggested by him.

Line 262: seems to be very heterogenic. Give SD or SE.#

Authors Reply: What was stated on this line were the minimum and maximum bulk density values, so it is impossible to provide an SD or SE for a minimum and maximum value.

Line 328: You explained that bulk densities increased. Even when this was observed with a high variation, soil carbon stocks in turn should be higher, if C content remained constant?!

Authors Reply: The reviewer is likely correct, but our data suggests that no change in SOC has occurred…..yet. In our manuscript, we do elude to the fact that increased beta-glucosidase activity could be an early indicator of other biological changes (Bandick and Dick, 1999) and likely would lead to increased SOC in the future.

Table2 and 3: Table 2 is for me personally more helpful as the used index. Actually, I have my doubts about the benefits of this used index to understand the presented results.

Authors Reply: We can see how table 2 would be preferred by most soil scientists (and others) who will read this manuscript. Why? Because you can see the actual change in soil indicators over time and depth. The Soil Management Assessment Framework takes the data from Table 2, in conjunction with other soil variables such as soil series, texture, clay content, climatic conditions, and uses preset algorithmic functions (based on more is better, less is better, somewhere in the middle is better concepts) to assign unitless scores from 0 (worst) to 1 (best), with the unitless scores shown in Table 3. For others that would want to follow a similar approach in the future with MiG (or other systems), we feel it is important to show the indictor score outcomes from the SMAF for individual indicators. It really helps tie the entire p

Please also note the supplement to this comment:
https://www.soil-discuss.net/soil-2019-91/soil-2019-91-AC2-supplement.pdf
* * *